# Chemerin: A Functional Adipokine in Reproductive Health and Diseases

**DOI:** 10.3390/biomedicines10081910

**Published:** 2022-08-07

**Authors:** Ming Yu, Yali Yang, Chen Huang, Lei Ge, Li Xue, Zhonglin Xiao, Tianxia Xiao, Huashan Zhao, Peigen Ren, Jian V. Zhang

**Affiliations:** 1Center for Energy Metabolism and Reproduction, Shenzhen Institute of Advanced Technology, Chinese Academy of Sciences, Shenzhen 518055, China; 2Shenzhen Institute of Advanced Technology, Chinese Academy of Sciences, Shenzhen 518055, China; 3Shenzhen Key Laboratory of Metabolic Health, Shenzhen 518055, China; 4Shenzhen College of Advanced Technology, University of Chinese Academy of Sciences, Shenzhen 518055, China

**Keywords:** chemerin, CMKLR1, GPR1, reproduction, pregnancy, PCOS, preeclampsia

## Abstract

As a multifaceted adipokine, chemerin has been found to perform functions vital for immunity, adiposity, and metabolism through its three known receptors (chemokine-like receptor 1, CMKLR1; G-protein-coupled receptor 1, GPR1; C-C motif chemokine receptor-like 2, CCRL2). Chemerin and the cognate receptors are also expressed in the hypothalamus, pituitary gland, testis, ovary, and placenta. Accumulating studies suggest that chemerin participates in normal reproduction and underlies the pathological mechanisms of certain reproductive system diseases, including polycystic ovary syndrome (PCOS), preeclampsia, and breast cancer. Herein, we present a comprehensive review of the roles of the chemerin system in multiple reproductive processes and human reproductive diseases, with a brief discussion and perspectives on future clinical applications.

## 1. Introduction

In eutherians, sexual reproduction is a highly complex and exquisite process that propagates genetic information to offspring. A functional reproductive system controlled by proper endocrine, metabolic, and immune factors is essential to building the pregnancy, which compromises continuous physiological events, including gametogenesis, fertilization, embryo development, embryo implantation, placentation, and parturition. The success of each step is requisite for the initiation of the next stage [1]. In humans, dysfunctional reproductive systems or disrupting each process of pregnancy could result in several reproductive-related disorders, such as polycystic ovary syndrome (PCOS) and preeclampsia.

The Industrial Revolution and modern civilization have brought remarkable convenience to our life and extend the average human life span [2]. Although the global population continues to grow, however, a grievous decline in total fertility rates (TFR) in many countries has occurred over the past decades [3]. Reproductive health problems in males (poor quality of sperm) and females (abnormal functions of ovary, oviduct, and endometrium) occurred frequently, and certain factors such as obesity (overnutrition), environmental pollution, and lifestyle were evidenced to link to these health issues [4]. Moreover, despite the application of assisted reproductive technologies that have substantially aided couples with reproductive disorders to have children, the gravidity success rates remain frustratingly low [5]. Therefore, understanding the molecular mechanisms underlying the physiopathological processes in reproductive events and related diseases holds the key to improving human reproductive health.

Worldwide increasing overweight and obesity are closely linked with a higher incidence of infertility in both sexes [6,7]. Maternal obesity also elevated the risk of gestational diabetes, pre-eclampsia, preterm birth, and metabolic diseases in offspring [8]. Additionally, robust evidence links obesity with 13 different types of cancer, including breast (postmenopausal), ovary, and endometrium cancer [9]. As the largest endocrine organ, adipose tissues are found to release plentiful factors, including lipids, metabolites, exosomal microRNAs, and bioactive peptides (termed “adipokines”), and these substances act through multiple organs to maintain physiological homeostasis [10]. In the context of reproductive research, clinical and experimental studies have revealed that certain adipokines (leptin, etc.) are responsible for regulating the reproductive systems and pregnancy. Abnormal circulating adipokines due to underweight or obesity cause reproductive-related diseases [11].

Chemerin is mainly secreted by adipose tissue and has been found to perform functions vital for immunity, adiposity, and metabolism through the receptors. Accumulating studies suggest that the chemerin system participates in normal reproduction and underlies the pathological mechanisms of certain reproductive system diseases, including polycystic ovary syndrome (PCOS),preeclampsia and gynecological cancer [12,13]. Herein, we present a comprehensive review of the roles of the chemerin system in multiple reproductive processes and reproductive-related diseases, with a brief discussion and perspectives on future clinical applications.

## 2. Chemerin

In 1997, a gene named tazarotene-induced gene 2 (*TIG2*) was first identified in psoriatic skin lesions [14]. The name was coined because its production is responding to tazarotene (a retinoic acid receptor beta/gamma-selective anti-psoriatic synthetic retinoid); additionally, this gene is also termed retinoic acid receptor responder 2 (*RARRES2*) [15]. In 2003, two independent groups reported that chemerin, the product of *TIG2*, was the endogenous ligand for an orphan G-protein coupled receptor (GPCR): chemokine-like receptor 1 (CMKLR1), also called chemerin receptor 1 or ChemR23 [16,17]. In the beginning, chemerin was mainly treated as a chemokine since it recruits the CMKLR1-expressing leukocytes to the inflammatory site to regulate the immune events [16]. However, literature published in 2007 reported that human and mouse adipocytes expressed high levels of chemerin and CMKLR1, and this signaling pathway regulated adipogenesis and adipocyte metabolism, suggesting that chemerin is a novel functional adipokine [18]. 

Chemerin is not specifically derived from fat deposits; like other adipokines, it is widely distributed in many organs. According to the human protein atlas (HPA) database, chemerin mRNA is highly expressed in endocrine tissues (adrenal gland, parathyroid, etc.), liver, pancreas, female reproductive system (ovary, cervix, endometrium, etc.), adipose tissue, lung, kidney, and colon. Moreover, high expression levels of chemerin mRNA are also observed in liver and adipose tissue of other species, such as mice, rats, and pigs [15]. 

Chemerin biology is complex, and one reason for this is that many protein isoforms with different bioactivity exist in the circulating or local environment. In human cells, *RARRES2* translates into a 163 amino-acids protein (pre-pro-chemerin). Subsequently, a 143 amino-acids protein (pro-chemerin, chemerin-S163) is secreted by cleavage of the N-terminal 20 amino-acids signal peptide. Pro-chemerin has low biological activity, and its C-terminal requires further proteolysis processing by various proteases to generate at least six isoforms, including chemerin-K158 (low activity), chemerin-S157 (highest activity), chemerin-F156 (high activity), chemerin-A155 (inactivity), chemerin-F154 (inactivity), and chemerin-G152 (inactivity) [15,19,20]; more details are shown in Figure 1A. Due to the critical role of the C-terminal proteolysis processing for chemerin activity, several synthetic peptides such as chemerin-9 [21], chemerin-13 [21], chemerin-15 [22,23], chemerin-20 [24], chemerin peptide analog CG34 [25,26] and cyclic peptide-9 [27,28] were found to exhibit the bioactivity in distinct model systems, indicating a potential application value of these peptides for experimental or clinical intervention. 

## 3. Chemerin Receptors

In addition to CMKLR1, two other GPCRs including G protein-coupled receptor 1 (GPR1, chemerin receptor 2) [29] and chemokine CC-motif receptor-like 2 (CCRL2, chemerin receptor 3) [30] were identified as the chemerin receptors. Based on the HPA database, CMKLR1 mRNA is highly expressed in various types of innate immune cells (plasmacytoid dendritic cells, macrophages, etc.), female reproductive system (placenta, endometrium, etc.), lung, muscle tissues, endocrine tissues, and adipose tissues. GPR1 mRNA is highly expressed in the brain (choroid plexus), esophagus, skin, placenta, adrenal gland, testis, ovary, gallbladder, and adipose tissues. CCRL2 mRNA is widely distributed and with the highest levels found in the lung, gastrointestinal tract, adipose tissues, breast, placenta, and immune cells (macrophages, etc.) [15].

The chemerin receptors display different characteristics in terms of ligand affinity, G protein activation, β-arrestin recruitment, calcium mobilization, and extracellular signal-regulated kinase (ERK) signaling activation. Briefly, although chemerin displays high affinities for all three receptors (K_D_: CMKLR1, 0.88 ± 0.33 nM; GPR1, 0.21 ± 0.02 nM; CCRL2, 2.35 ± 1.23 nM), only CMKLR1 activates the Gαi and Gαo proteins, whereas GPR1 and CCRL2 hardly activate any G proteins [31]. CMKLR1 and GPR1, but not CCRL2, could result in the β-arrestins recruitment and receptor internalization upon the stimulation of chemerin or chemerin-9 [31]. In primary macrophages, chemerin-mediated CMKLR1 activation recruits the G protein-coupled receptor kinase 6 (GRK6) [32], which phosphorylates the intracellular domains of CMKLR1 thus terminating the signaling pathways by desensitization or internalization. Furthermore, the phosphorylation sites at Ser343 (predicted for GRKs) and Ser347 (predicted for protein kinase C) are essential for the internalization of CMKLR1 induced by chemerin or chemerin-9, and this internalized pattern might be regulated through caveolae-mediated endocytosis instead of clathrin-coated pits [33]. More importantly, CMKLR1 displays a strong signals response to chemerin (intracellular Ca^2+^ release, suppression of cAMP accumulation, and phosphorylation of ERK1/2). Unlike CMKLR1, the binding of chemerin to GPR1 results in a weak Ca^2+^ mobilization and phosphorylation of ERK1/2, and the binding of chemerin to CCRL2 does not signal nor internalize [31]. Recent findings reported that GPR1 displayed rapid ligand-independent, constitutive internalization and acted as a scavenging receptor with wider ligand specificity [34,35]. These results indicate that CMKLR1 is a typical GPCR for chemerin with canonical G protein-related signaling features, while GPR1 and CCRL2 are atypical GPCRs (Figure 1B), and more detailed properties of these receptors are needed for further work.

## 4. Roles of the Chemerin System in Reproduction

### 4.1. Roles of Chemerin System in Hypothalamus–Pituitary–Gonadal (HPG) Axis

#### 4.1.1. Hypothalamus and Pituitary Gland

In mammals, reproductive functions are controlled by an intricate and highly coordinated neuroendocrine axis, which is termed the hypothalamus–pituitary–gonadal (HPG) axis. The specialized hypothalamic neurons synthesize gonadotropin-releasing hormone (GnRH), which regulates the pituitary glands to produce luteinizing hormone (LH) and follicle-stimulating hormone (FSH). These two gonadotropins are the master endocrine mediators of gonadal functions, including gametogenesis and steroidogenesis, and the sex hormones produced by gonads further control the hypothalamic GnRH neurons in a negative feedback manner [36]. Dysfunctional HPG axis due to endocrine abnormality or metabolic disorder critically affects the sex behaviors and reproductive functions. 

The chemerin system is found to be expressed in the hypothalamus and pituitary gland, implying their potential role in controlling the reproductive neuroendocrine events [37]. Chemerin transcripts exist in the hypothalamus of baboons and chimpanzees [38] and are restricted to the dorsal ventricular wall of the anterior, medial, and posterior hypothalamus in mice [39]. In rats, chemerin transcripts are localized in the ependymal cells and tanycytes lining the third ventricle and stalk-median eminence (SME) region of the hypothalamus, and CMKLR1 transcripts are found in the prefrontal cortex, hippocampus, cerebellum, ependymal cell layer and SME [40,41]. CCRL2 transcripts are also localized in the ependymal cell layer and SME of rats [40]. On the other hand, GPR1 protein is expressed in the hypothalamus of female mice, and colocalized with GnRH- and corticotropin-releasing factor (CRF)-positive cells, indicating that GPR1 may be involved in the HPG axis and hypothalamus–pituitary–adrenal (HPA) axis [42]. In pigs, chemerin and its three receptors are detected to express in the hypothalamic structures responsible for GnRH biosynthesis and release [43]. Chemerin is also observed to express in the tissues of porcine pituitaries and colocalized with LH- and FSH-positive cells [44].

Chemerin (circulating or locally produced in the brain) appears to influence the release of gonadotropins in the pituitary gland. Evidence showed that serum chemerin levels were lower in subfertile men relative to normal individuals and inversely associated with LH, E2, and sex-hormone binding globuline (SHBG). Chemerin levels were lower in men with high LH levels than in men with normal LH levels [45]. Additionally, recombinant chemerin (r-chemerin) influenced both basal, GnRH- and/or insulin-mediated LH and FSH in porcine primary anterior pituitary cells [44]. In female mice, deficiency of *Gpr1* leads to lower mRNA values of GnRH in the hypothalamus, and higher mRNA values of FSH in the pituitary glands, along with higher serum E_2_ levels relative to the wild type [46]. However, deficiency of *Cmklr1* does not affect the serum LH, FSH, estradiol (E_2_), and progesterone (P_4_) levels in female mice [47]. These results indicate that GPR1, but not CMKLR1 probably takes part in the regulation of the hypothalamus–pituitary–ovarian (HPO) axis.

#### 4.1.2. Testis

The regulation of male reproductive functions mainly relies on the hypothalamus–pituitary–testicular (HPT) axis. LH stimulates the testicular interstitial cells (Leydig cells) to synthesize and secrete testosterone (T), which is essential for spermatogenesis in seminiferous tubules of the testis. FSH acts on the testicular Sertoli cell to proliferate in prepuberty and determines the size of the testes as well as amplifies the spermatogenesis-promoting effects of T. FSH cooperates with T, resists the apoptosis of the male germ cells, and elevates sperm production [48,49].

Chemerin, CMKLR1, and GPR1 protein were reported to be concentrated in the Leydig cells of human and rat testes. Furthermore, r-chemerin impeded the human chorionic gonadotropin (hCG)-stimulated T production in primary murine Leydig cells, which was accompanied by the declined phosphorylation of ERK1/2, and the downregulation of 3β-hydroxysteroid dehydrogenase (3β-HSD) [50]. Similar results have been verified in chicken testis explants, and the r-chemerin impaired the quality and motility of sperm in roosters, and this effect was antagonized when spermatozoa were pre-treated with CMKLR1 antibodies [51]. In addition, *Cmklr1* knockout in male mice leads to a declined number of Leydig cells in the testis [52], and *Cmklr1*^−/−^ and *Gpr1*^−/−^ male mice exhibit lower serum T levels when compared to the wild-type animals [52,53]. On the other hand, Brzoskwinia et al. found that flutamide (an anti-androgen drug) administration inhibited the chemerin, CMKLR1, and GPR1 expression in rat Leydig cells relative to the control groups [54]. These data implicate a feedback loop of the androgen/chemerin axis in Leydig cells, and other biological functions of chemerin and its receptors in the process of spermatogenesis remain to be demonstrated.

#### 4.1.3. Ovary

In the female, FSH promotes follicular growth through stimulating the mitosis of granulosa cells, accompanied by the estrogen biosynthesis in these cells. LH drives oocyte maturation, ovulation, and the formation of corpus luteum (progesterone biosynthesis), which is developed from the residual follicular granulosa and thecal cells (luteal cells) after ovulation. If pregnancy does not occur, uterine-derived prostaglandin F2α (PGF2α) compels the corpus luteum to regress (luteolysis); subsequently, a new reproductive cycle begins again [55,56].

Chemerin and CMKLR1 are expressed in the human ovary and especially in granulosa cells, and chemerin levels are at least two times higher in follicular fluid than in plasma [57]. In the mouse ovary, GPR1 protein is localized to thecal cells, granulosa cells, luteal cells, and stromal cells [58]. Furthermore, chemerin and its receptors are also found in the ovaries of rats [59], pigs [60], bovine [61], hens [62], and turkeys [63]. Thus, the chemerin system appears to participate in controlling female reproduction at the ovarian level. 

As reported originally by Reverchon et al., chemerin has an inhibitory effect on ovarian steroidogenesis. In human primary granulosa cells, r-chemerin treatment suppressed insulin-like growth factor-1 (IGF-1)-induced P_4_ and E_2_, but did not affect basal- or FSH-induced steroid production. Mechanistically, r-chemerin reduced IGF-1-induced aromatase expression and cell proliferation by inactivating the IGF-1R signaling pathways [57]. Similar inhibitory effects of chemerin on ovarian steroidogenesis were observed in mice (hCG-induced P_4_) [58], rats (FSH-induced P_4_ and E_2_) [64] and bovine (basal-, FSH-, and IGF-1-induced P_4_ and E_2_) [61]. In pigs, chemerin also affects ovarian steroidogenesis, whereas it exhibits varied effects (accelerative and inhibitory) on the secretion of P_4_, androstenedione (A_4_), T, estrone (E_1_) and E_2_ [65]. Meanwhile, the locally enhanced chemerin with bioactivity may negatively affect follicular development and oocyte maturation. For instance, r-chemerin treatment of bovine cumulus–oocyte complexes (COCs) resulted in the meiotic arrest and impeded bovine oocyte nuclear maturation [61]. Furthermore, in high-fat diet (HFD) mice, ovarian chemerin and CMKLR1 expression were enhanced, and the chemerin/CMKLR1 signaling pathway contributed to reactive oxygen species (ROS) accumulation and apoptosis through AKT, AMPK, and NF-κB pathways in mouse granulosa cells [66]. 

Chemerin/GPR1 signaling pathway was evidenced to regulate P_4_ generation during follicular development and corpus luteum formation as well as PGF2α-induced luteolysis in mice [58]. In porcine luteal cells, r-chemerin treatment altered the expression of biomarkers related to angiogenesis and apoptosis [67]. The regulatory effects of chemerin on corpus luteum in pigs were further verified by transcriptomic analysis, which showed that chemerin takes part in the processes including steroids and prostaglandins synthesis, inflammatory response as well as luteotropic and luteolytic signals [68,69].

### 4.2. Roles of Chemerin System in The Endometrium

In viviparous mammals, embryo implantation is the first stage for establishing direct contact between the mother and the fetus. It requires the synergistic interaction between a receptive endometrium and a capable embryo and occurs at a limited time called the “implantation window” [1]. The human endometrium is a sex hormones-targeted tissue that in the absence of successful implantation sheds each month. After the menstrual phase, the endometrium continues to proliferate and rebuild under the control of estrogen. In this period, also called the proliferative phase (equal to the follicular phase, FP), the endometrium is non-receptivity. Following ovulation, the corpus luteum secretes progesterone, which antagonizes the estrogen-induced cell proliferation, and gradually transforms the endometrium into a receptive status. This stage is termed the secretory phase (equal to the luteal phase, LP), and during the mid-secretory phase (equal to the mid-luteal phase, mid-LP), the endometrial receptivity reaches the highest, waiting for the blastocyst [70,71].

In healthy eumenorrheic women, there is no significant change in serum chemerin concentration between the early follicular phase and the mid-luteal phase [72], which indicates that circulating chemerin seems not responsive to the hormone fluctuations throughout the menstrual cycle in humans. Limited studies reported the expression of the chemerin system in human or rodent endometrium, and whether it plays a key role in endometrial receptivity in these species is still unknown. However, in pigs, chemerin and GPR1 protein levels were found to reach the highest in the endometrium during the mid-luteal phases. CMKLR1 and CCRL2 protein levels were shown to reach the highest in porcine endometrium derived from the late-LP and FP, respectively. In uterine luminal flushings (ULF), chemerin levels were found to significantly fluctuate throughout the estrous cycle, with the highest level during the LP [73]. These findings suggest that the expression of chemerin and its receptors in the porcine endometrium responded to the fluctuant hormonal milieu. More importantly, transcriptomic and proteomic analyses further showed that recombinant chemerin affected many genes and proteins related to critical events of implantation (proliferation, adhesion, migration, invasion, angiogenesis, immune response, etc.) in porcine endometrial explants derived from the peri-implantation period [74,75]. These results provide the molecular basis that links the chemerin axis with endometrial receptivity establishment, and more studies are warranted in human or murine endometrial cells. 

### 4.3. Roles of Chemerin System in Placentation

The placenta is a transient organ, that sits at the interface between the mother and fetus, and fulfills the tasks ranging from exchange of nutrients and waste, regulation of immunological acceptance, and support of fetal development [76]. At the time of implantation (7–9 days post-fertilization), the human blastocyst is segregated into two lineages: the inner cell mass (ICM) and trophectoderm (TE). Blastocyst adheres to the endometrial luminal epithelium, triggers the TE fuses to form an invasive primitive syncytium, and thereafter penetrates into the endometrial decidua. Next, the placental villi are formed, and this basic structure of the definitive placenta compromises the fetal blood vessel that is encompassed by villous trophoblasts (outer layer: syncytiotrophoblasts, STBs; inner layer: cytotrophoblasts, CTBs) [77]. At the earlier stage of the first trimester, CTBs either fused to form the multinucleated STBs or give rise to the extravillous trophoblasts (EVTs) with a highly invasive phenotype, which are further embedded into the vascular wall of spiral arteries to assist the establishment of maternal blood supply (10–12 weeks of pregnancy) [78]. During the second and third trimesters, the placenta is fully developed and both the mother and the fetus continue to grow. 

In normal pregnant women, circulating chemerin concentrations significantly rise as pregnancy progresses [72,79], and are negatively associated with adiponectin (an insulin-sensitizing adipokine) levels but not correlated with the markers of metabolism and insulin sensitivity [72]. In addition, chemerin is positively correlated with body mass index [79], suggesting that the origin of gestational chemerin is likely derived from the maternal adipose tissue. In pregnant rats, the serum chemerin levels gradually decrease [80]. In pigs, high levels of chemerin in ULF were found during early pregnancy, with the highest levels on the pregnant days (PDs) 12–13, lower on PDs 15–16, and the lowest on PDs 10–11 and PDs 27–28 [73]. By comparing the primary human stromal cells (STs), EVTs, and decidual endothelial cells (DECs) derived from the first-trimester decidual tissues, chemerin mRNA and protein were observed to be highly expressed in STs and EVTs, but not DECs [81]. Additionally, chemerin protein was also observed in CTBs, Hofbauer cells, and vascular endothelial cells of the human placenta during the third trimester [80]. In rats, placental chemerin mRNA levels rise significantly at PD16 and declined gradually towards the end of the pregnancy, and chemerin protein is detected to localize in both the labyrinthine trophoblast and trophospongium of the placenta at PD19 [80]. CMKLR1 protein was detected in decidual natural killer (dNK) cells and chorionic villi derived from the human decidual tissues in the late first trimester. In pigs, the highest protein levels of chemerin and its receptors were observed in the conceptuses and trophoblasts (except for GPR1) during placentation [73].

The role of the chemerin/CMKLR signaling pathway in placentation has begun to emerge. Recently, r-chemerin was reported to promote the syncytialization of CTBs by using the in vitro model of human choriocarcinoma BeWo cells [82]. The r-chemerin also inhibited the proliferation of dNK cells in vitro, and the conditioned medium derived from the dNK cells pre-treated with r-chemerin significantly impaired the invasion of EVTs in vitro [82]. Meanwhile, chemerin secretion was found to be enhanced in decidualized human endometrial stromal cells, and chemerin positively regulated the chemotaxis of NK cells and angiogenesis of HUVEC cells in vitro [81]. More importantly, *Cmklr1* knockout led to impaired functional placental labyrinth development including reduced fetal vessel density and STBs differentiation. *Cmklr1*^−/−^ mice exhibited an enlarged diameter of the spiral arteries, enhanced trophoblast invasion, and an increased number of dNK cells in the decidual zone. The dysfunctional placentation due to *Cmklr1* knockout ultimately affected the fetus’s growth and development [82]. These findings are also supported by an article, which reported that injection of 2-(α-naphthoyl) ethyltrimethylammonium iodide (α-NETA, an antagonist for CMKLR1) in mouse uterine horn during the early pregnancy resulted in the abortion of embryos [83]. Combining these data, the chemerin/CMKLR1 singling pathway plays a critical role in placentation during early pregnancy, and more details need further investigation.

## 5. Roles of Chemerin System in Reproductive System Diseases

### 5.1. Polycystic Ovary Syndrome

Polycystic ovary syndrome (PCOS) is the most common endocrine and metabolic disorder in women, affecting roughly 15% of women of childbearing age [84]. This heterogeneous disease is divided into four phenotypes depending on the presence or absence of three characteristics: hyperandrogenemia, ovulatory dysfunction, and polycystic ovarian morphology [85]. PCOS patients usually display basal and glucose-stimulated hyperinsulinemia and insulin resistance (IR). Besides the metabolic dysfunction, PCOS undoubtedly causes subfertility and also increases the long-term risks for other diseases such as type 2 diabetes mellitus (T2DM) and ovarian cancer [85]. The intricated etiology of PCOS is probably attributed to the combination of genetic, epigenetic, metabolic, endocrine, and immune factors. Current therapeutics for this muti-phenotypic disease are suboptimal, and a universal and specific drug for PCOS treatment is sorely needed [81].

Chemerin expression was found to be significantly increased in ovarian tissues [86], granulosa-lutein cells [87], and adipose tissues [88] in women with PCOS patients relative to normal individuals. CMKLR1 expression was also up-regulated in granulosa-lutein cells from women with PCOS relative to the healthy subjects [87,89,90]. There is extensive literature reporting that chemerin levels in blood samples and follicular fluids (FF) were significantly enhanced in PCOS patients relative to healthy women [86,91,92,93,94]. Serum chemerin values were correlated with other adipokines values (leptin, r = 0.508; adiponectin, r = −0.36; leptin/adiponectin ratio, r = 0.605) [95], and positively correlated with BMI [95], the incidence of abortion [96], and assisted reproductive failure [97] in PCOS patients. Elevated FF-chemerin seems to be a risk factor for a decrease in oocyte utilization rate and high-quality embryo rate [98]. These data point out that chemerin detection in serum or FF offers a potential diagnostic value for PCOS development and pregnancy outcomes in PCOS patients.

Chemerin has been established to play an essential role in glucose homeostasis [99]. Insulin infusion in healthy human subjects significantly augmented serum chemerin levels, and after 6 months of metformin treatment in PCOS patients, there were obvious declines in serum chemerin levels, waist-to-hip ratio (WHR), E_2_, T, glucose, and homeostatic model assessment of insulin resistance (HOMA-IR) [88]. Similar results were observed in PCOS patients after 3 months of metformin therapy [100]. In parallel, insulin promoted the expression of chemerin in human granulosa-lutein cells (hGLs) [87] and human omental adipose tissue explants (hOATEs), while metformin significantly decreased the expression of chemerin in hOATEs [88]. Chemerin levels were significantly elevated in both follicular fluid and hGLs from PCOS patients with IR compared with the PCOS patients without IR. Chemerin treatment attenuated insulin-induced glucose uptake in hGLs, and this effect was abolished by CMKLR1 knockdown via siRNA interference [87]. These studies imply that: 1. Serum chemerin measurement may be of value in PCOS after certain treatments, especially metformin; 2. The chemerin/CMKLR1 axis is a potential target for the improvement of IR in PCOS.

Rodents have been utilized to create PCOS-like models to increase our molecular understanding of this disease [101]. The chemerin and CMKLR1 expression in antral follicles were found to increase in the dihydrotestosterone (DHT)-induced PCOS model [59,102], and exogenous r-chemerin inhibited follicular growth and induced apoptosis of granulosa cells in vitro [59]. Meanwhile, r-chemerin suppressed FSH-induced aromatase and cytochrome P450scc expression in preantral follicles and granulosa cells, which further compromised the E_2_ and P_4_ secretion, respectively [102]. Moreover, *Cmklr1* deficiency partially alleviated the ovarian functions (ovary morphology and steroidogenesis) and lipid accumulation caused by DHT treatment in female mice [47,103]. *Gpr1* deficiency also alleviated the DHEA-induced weight gain and ovarian morphological changes, improved the expression of steroid enzymes in ovaries, and E_2_ synthesis in cultured granulosa cells, and these effects were partially through the mTOR signaling pathway [46]. More importantly, specific nanobody targeting CMKLR1 abolished chemerin-induced P_4_ inhibition in human luteinized granulosa cells [89], and antibodies targeting GPR1 attenuated DHEA-induced E_2_ in mouse ovarian granulosa cells [46]. These studies demonstrate that the chemerin system participates in the pathogenesis of PCOS, and targeting the receptors provides a promising therapeutic strategy.

Chronic inflammation has been recognized as one of the major causes or consequences of PCOS, as supported by the alterations of inflammatory factors/cells in circulating and ovarian local in PCOS patients [104]. Chemerin certainly plays a vital role in inflammation, as CMKLR1 is highly located on the surface of certain immune cells. By using the DHT-induced PCOS rat model, Lima et al. found that DHT changes the balance of ovarian M1 (inflammatory) and M2 (anti-inflammatory) macrophages, which are responsible for the granulosa cell apoptosis. Enhanced ovarian chemerin by hyperandrogenemia seems to recruit the CMKLR1-positive macrophages to influence the immune microenvironment of ovaries [105]. This study established an immunological perspective on PCOS at the ovarian level by linking the ovary-derived chemerin with the inflammation, and more studies in this research direction are required.

### 5.2. Endometriosis and Endometritis

Endometriosis (EM) is a hormone-dependent and chronic inflammatory disease defined by the presence of endometrial lesions outside the uterus, which is estimated to affect 10% of reproductive-age women. It could be divided into three sub-types based on its histopathology and anatomical locations: superficial peritoneal endometriosis, ovarian endometriosis (also known as endometrioma or chocolate cyst), and deep infiltrating endometriosis (DIE) (nodules > 5 mm in depth). This debilitating disease usually causes chronic pelvic pain and infertility, and currently there are suboptimal diagnosis and treatment options [106]. In 2015, literature reported that: 1. Chemerin contents in peritoneal fluids (PF) from patients with endometriosis were significantly enhanced when compared with the controls; 2. Chemerin levels in PF were positively correlated with both TNF-α and IL-6 levels in PF. 3. Chemerin and CMKLR1 expression were significantly up-regulated in the ovarian endometrioma tissues relative to the eutopic endometrial tissues [107]. This preliminary study suggests that the chemerin/CMKLR1 signaling pathway is aberrant in endometriosis, but the fundamental mechanisms remain largely uncharacterized.

As another inflammatory disease of the endometrium, endometritis impairs the ovarian and endometrial functions, which in turn causes poor reproductive outcomes [108]. By comparing healthy, transient, and persistent endometritis cows, chemerin concentrations in uterine fluid (UF) from the persistent groups were significantly higher than in the healthy and transient groups and allowed for this distinction in 100% of cases. Cows with persistent endometritis displayed increased transcript levels of chemerin, CMKLR1, and GPR1 in the cellular pellet of UF and an up-regulated protein amounts of chemerin and CMKLR1 in the endometrium. This study implicates that chemerin represents a suitable biomarker for cytological endometritis in dairy cows [109], and chemerin systems might involve in the inflammatory activities of the uterus in humans.

### 5.3. Preeclampsia

Preeclampsia (PE) is a common pregnancy-related disease, which is estimated to affect 4.5% of gestational women. It is characterized by new-onset hypertension, edema, and proteinuria that develop after 20 weeks of gestation in previously normotensive women [110]. The pathophysiology of PE is intricated, and it appears to progress in two stages: 1. Abnormal placentation in the first trimester due to the insufficient trophoblast invasion, which gives rise to poor spiral artery remodeling and ischemia-reperfusion injury; 2. An increasingly stressed placenta causes the clinical signs of PE, including hypertension, proteinuria, clotting, and liver dysfunction in the second and third trimesters [111]. Meanwhile, maternal obesity represents a risk factor for the progression of PE [112], and adipokines may be involved in the pathogenesis of PE.

Accumulating studies have shown that chemerin amounts in serum are significantly elevated in PE patients [113,114,115,116], even after delivery for 6 months [115], and were closely associated with the disease severity [113,114,116] and adverse neonatal outcomes [114]. Chemerin level of >252.0 ng/mL in maternal serum predicted PE with high sensitivity and specificity [114]. The ROC curve analysis from 518 pregnant women points out that serum chemerin measurement in the first trimester provides a high predictive value for PE [113]. Chemerin levels are also positively associated with several clinical parameters including systolic and diastolic blood pressure, HOMA-IR, AST, ALT, proteinuria, and the markers of dyslipidemia [114,116].

Chemerin was predominantly located in the spongiotrophoblast and giant cells of the placenta-derived from the intrauterine growth restriction (IUGR)-like rat model and PE-like mouse model [117]. Increased chemerin and CMKLR1 protein levels were also found in the placenta derived from the PE-like rat model [118] and human patients [119]. The chemerin/CMKLR1/AMPK/TXNIP/NLRP3 axis promoted the pyroptosis and inflammation of human HTR8/SVneo trophoblast cells and rat primary trophoblast cells in vitro, indicating chemerin serves as a pro-inflammatory inducer to aggravate the progression of PE [118]. Additionally, chemerin stimulated M1 macrophage polarization through the CMKLR1/Akt/CEBPα/IRF8 signaling axis, thus inhibiting macrophage-induced trophoblast invasion and migration as well as suppressing macrophage-mediated angiogenesis in vitro. When blockage of the chemerin/CMKLR1 axis by α-NETA occurs, M1 macrophage polarization and PE-like syndromes are both alleviated in a rat PE model [120]. Recently, a pregnant mouse model with trophoblastic-specific overexpression of chemerin was reported to manifest a PE-like phenotype (hypertension, proteinuria, and endotheliosis) accompanied by compromised trophoblast invasion, up-regulation of sFlt-1 (soluble Fms-like tyrosine kinase-1) and the inflammation markers as well as the poor pregnancy outcomes [119]. Combining these data, targeting the chemerin/CMKLR1 signaling pathway appears a plausible therapeutic option for PE treatment. However, chemerin may play a protective role by regulating nitric oxide signaling in vascular cells in PE [121], and future studies should assess the relative contribution of chemerin on trophoblasts, immune cells, and vascular cells in PE. 

### 5.4. Gestational Diabetes Mellitus

Gestational diabetes mellitus (GDM) is defined as glucose intolerance with onset or first detection during pregnancy, which influences approximately 14% of pregnancies worldwide [12]. Following a diabetic pregnancy, both mother and newborn are at an increased risk for metabolic and cardiovascular disease, and studies have identified several GDM risk factors, including maternal overweight or obesity, age, ethnicity, previous history of GDM, and family history of T2DM [122]. In normal pregnancy, physiological IR occurs probably due to the placental hormones, although the mechanism remains unclear [123]. In pregnant women with normal glucose tolerance, IR triggers the pancreatic β cells to release insulin to sustain euglycemia. Dysfunction of this response results in maternal hyperglycemia; accordingly, IR and abnormal β-cells underlie the pathophysiology of GDM [122,123].

In normal pregnant women, circulating chemerin amounts significantly rise as pregnancy progresses [72], which implies the possible relationship between chemerin and IR. However, the evaluation of correlations between chemerin, fasting insulin, and HOMA-IR values provided contradictory results [124]. Additionally, detection of chemerin levels in blood samples, placenta, and adipose tissues from GDM patients also presented conflicting results as summarized in greater detail by Gutaj et al. [124].

Although clinical and animal experimental studies have demonstrated the critical role of chemerin [99,125,126], CMKLR1 [127], GPR1 [128], and CCRL2 [129] in controlling the glucose homeostasis under a non-pregnant status, there is limited knowledge regarding the role of the chemerin system in GDM. Huang et al. observed that chemerin expression was increased, whereas GPR1 expression was decreased in the placenta of GDM-like mice and GDM patients. Knockdown of GPR1 by RNA interference aggravated glucose intolerance, hindered lipid metabolism, and impaired β-cell viability in GDM-like mice. More importantly, the silencing of GPR1 impaired the metabolism of glucose and lipids in placental trophoblasts as detected by the down-regulation of glucose transporter 3 (GLUT3) and fatty acid-binding protein 4 (FABP4). These results suggest that the engagement of the chemerin/GPR1 signaling pathway in regulating the metabolism during pregnancy and targeting GPR1 could be a promising approach for GDM treatment [130].

In the aggregate, despite some studies suggesting the predictive value of chemerin for GDM development, clinical data gathered at present are controversial and insufficient for diagnostic application. Additional animal experiments will be needed to identify the role of the chemerin system in GDM and also to dissect the molecular basis in detail.

### 5.5. Gynecologic Cancer

Obesity has been conclusively linked to three common types of gynecologic cancers (GCs): breast (postmenopausal), ovary, and endometrial cancer. However, the role of adipokines, particularly chemerin in tumorigenesis and progression is yet to be deciphered. As reviewed in greater detail by Treeck et al. [13], and Goralski et al. [131], RARRES2 expression is decreased, whereas the chemerin protein abundance is increased in breast cancer tissues compared with the normal samples, raising the question as to what causes the inverse trend of chemerin mRNA and protein expression in breast cancer. The data gathered at present have provided controversial results. Clinical information analysis suggested a potential pro-tumorigenic effect of chemerin in breast cancer, as demonstrated by the fact that the overall survival (OS) rate of patients expressing higher levels of chemerin was worse than that of those expressing lower levels [132]. A recent study suggested that elevated serum chemerin was positively correlated with Ki67 amounts in breast cancer tissues and histologic grade. Combing chemerin with CA15-3 resulted in a better diagnosis of breast cancer [133]. However, forced overexpression of chemerin by murine EMT6 cells significantly suppressed tumor growth by recruiting NK cells or CD8+ T cells in the orthotopic breast carcinoma model [134]. In parallel, recombinant chemerin treatment suppressed tumor growth in an MCF-7 cell-derived xenograft model and reduced the progression of osteolytic lesions in an MDA-MB-231 cell-induced osteolysis model [135]. On the other side, GPR1 was discovered to elevate in triple-negative breast cancer (TNBC) specimens and cell lines, and GPR1 binding peptide screened by phage display technology exhibited antineoplastic effects against TNBC in vitro and in vivo [136].

Despite high levels of bioactive chemerin being found to exist in ascitic fluids of patients with ovarian carcinomas, to date, few studies have addressed its impact on the pathophysiology of ovarian cancer. As reviewed in greater detail by Treeck et al. [13], an analysis of microarray data from 1656 ovarian cancer patients showed that RARRES2, CMKLR1, and GPR1 amounts were lower in ovarian cancer tissue than in the normal one, and CCRL2 values were comparable in normal and cancerous tissue. RARRES2 levels negatively, whereas CMKLR1 levels positively influenced both OS and progression-free survival (PFS) of ovarian cancer patients, and GPR1 and CCRL2 levels did not affect patients’ OS or PFS. Exogeneous chemerin treatment did not have any effect on the proliferation of HOSEpiC (normal), OVCAR-3 (ovarian epithelial carcinoma), COV434 (granulosa carcinoma), and KGN (granulosa carcinoma) cells in vitro [57,137]. While chemerin treatment promoted the proliferation and migration of HP8910 (ovarian epithelial carcinoma) cells by upregulating the expression of PD-L1 [138]. In contrast, knockdown of chemerin in COV434 cells was proved to inhibit cell proliferation and trigger apoptosis in vitro [139]. These conflicting results are likely due to the characteristics varied in different ovarian cancer cell lines or the diverse research strategies (exogenous recombinant protein treatment/endogenous gene manipulation; in vitro model/in vivo model), and more experimental experiments are required.

In sum, to date, the role of the chemerin system in breast, ovarian and endometrial cancer is debatable and limited. Chemerin has been shown to modulate various biological events, such as adipogenesis, differentiation, inflammation, and angiogenesis; thus, the contribution of chemerin and the cognate receptors on cancer cells and other cell components (adipocytes, fibroblasts, immune cells, and vascular cells) in tumor microenvironment will be needed to deeply dissected.

## 6. Discussion

Over the past two decades, the importance of chemerin in physiological events and diseases has been evidenced by numerous clinical and animal experimental studies [140]. Breakthrough discoveries provide insights that chemerin links the communication of cancer cells–immune cells [141], adipocytes–vascular cells [142], adipocytes–immune cells [143], and adipocytes–cancer cells [144]. However, the definite role of the chemerin system in human health and disorders is still not fully understood, especially in the fields of reproductive biology, gynecology, and obstetrics. Normal chemerin signals appear to regulate the HPG axis and endometrial receptivity as well as maintain the placentation during the early pregnancy (Figure 2), whereas an aberrant chemerin system due to obesity or inflammation may cause PCOS, EM, PE, GDM, and GCs (Figure 3). A major question is what is the relative influence of circulating chemerin (derived from the liver or adipose) and local chemerin (derived from the reproductive organs) on these reproductive events and related disorders. To this end, further research should be undertaken to investigate the phenotype and molecular basis by utilizing the animal models of conditional knockout or specific over-expression for chemerin in the particular organs.

Chemerin was evidenced as a valuable biomarker for diagnosis in a wide range of diseases, including obesity [145], cardiovascular diseases [146], T2DM [146], male subfertility [45], PCOS [86], and cytomegalovirus-induced intrauterine infection [147]. However, researchers and physicians should dissect the differential patterns of chemerin isoforms varied in diverse physiopathological microenvironments by using the mass spectrometry analysis or antibodies against specific chemerin isoforms [148]. For example, chemerin-K158 is the primary isoform in synovial fluids (derived from the patients with arthritis) and cerebrospinal fluids (derived from the patients with central nervous system diseases) but not plasma [149]. Chemerin-F156 and S157 increase in serum and follicular fluid derived from the patients with PCOS, while chemerin-A155, F156, and S157 enhance in serum derived from the patients with rheumatoid arthritis [150]. Accordingly, further studies are necessary to identify the specific composite patterns of chemerin forms in patients’ biofluids, especially the reproductive system diseases with poor diagnostic methods, such as unexplained infertility, preeclampsia, and endometriosis. 

In view of the aberrant expression of the chemerin system in various diseases, blockage or activation of the chemerin system confers a promising therapeutic approach. However, it is worth noting that the chemerin/CMKLR1 signaling pathway also regulates the normal physiological functions, such as metabolism and immune response [151]. Targeting the chemerin system should be carefully considered dependent on specific disease states and cellular context, and also distinguish the intracellular signaling pathways among the three receptors (CMKLR1, GPR1, and CCRL2) in detail. Currently, antibodies for chemerin neutralization [152], synthetic chemerin peptide [23], small-molecule antagonists for CMKLR1 (α-NETA [153] and CCX832 [154]), and nanobodies for CMKLR1 [155] were developed, and these reagents were shown to display effective therapeutic effects in several animal disease models. The biological role of GPR1 and CCRL2 remains largely unknown, and no small-molecule antagonists targeting these two receptors have been found until now. By utilizing the phage display techniques, our group screened a series of peptides binding with the CMKLR1 (unpublished study) or GPR1 [136]. The GPR1-binding peptide was evidenced to inhibit the triple-negative breast carcinoma [136] and improve LPS-Induced depressant-like behaviors and ovarian functions [42] in vivo. Further work is required to exploit the novel tools (compounds, nanobodies, etc.) to specifically target the diverse chemerin isoforms and the receptors. Additionally, optimizing the affinity, safety, stability, and half-time period in vivo of these potential “drugs” is also required for pharmaceuticals and clinic treatment in the future.

## 7. Conclusions

In conclusion, the information gathered in the last two decades certainly highlights that chemerin locates at the crossroads of inflammation and metabolism. However, research on this multifaceted adipokine in the context of reproductive biology remains in the primary stage, and more work is required to elucidate the precise role and underlying mechanism of the chemerin system in reproductive events and related disorders.

## Figures and Tables

**Figure 1 biomedicines-10-01910-f001:**
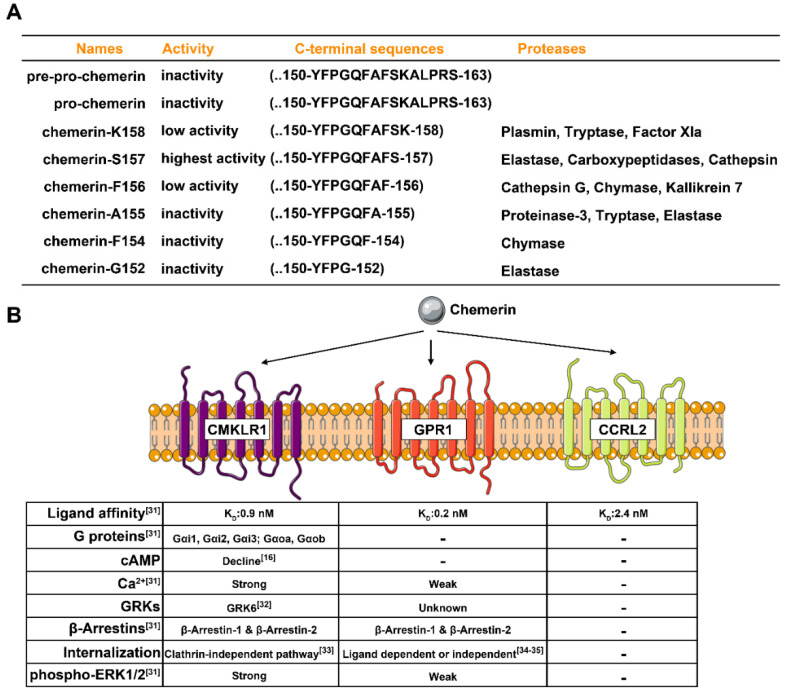
The characteristics of chemerin isoforms and the three receptors. (**A**) The characteristics of known chemerin isoforms are summarized based on Buechler et al. [19] and Ernst et al. [20]. (**B**) The known characters of CMKLR1, GPR1, and CCRL2 in terms of chemerin affinity and certain GPCR signaling features are summarized.

**Figure 2 biomedicines-10-01910-f002:**
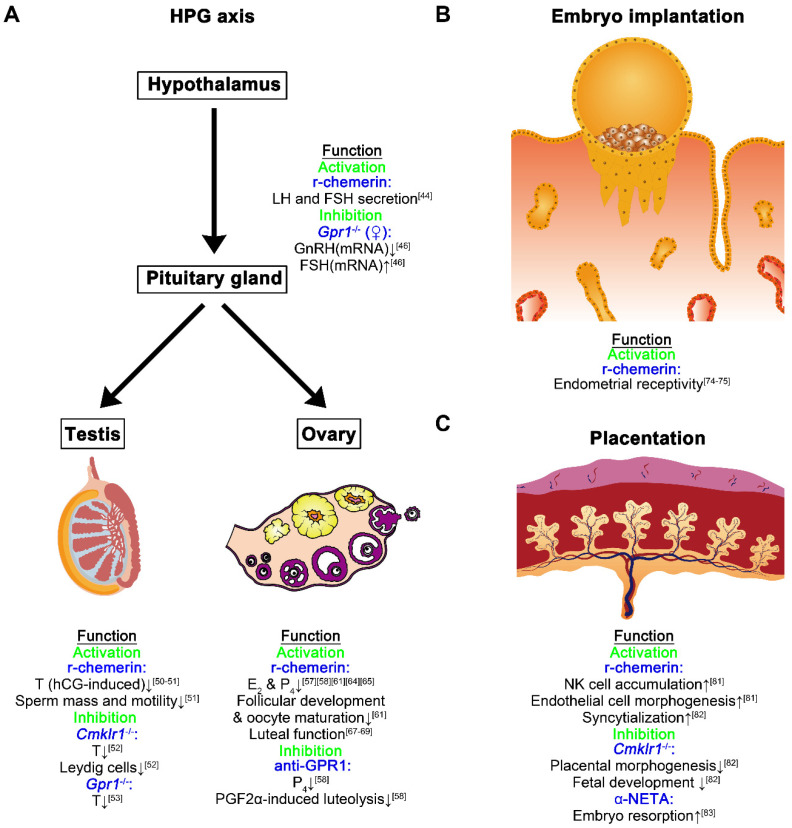
The biological functions of the chemerin system in physiological reproductive processes. The emerging information about the roles of the chemerin system in the HPG axis (**A**), embryo implantation (**B**), and placentation (**C**). r-chemerin, recombinant chemerin.

**Figure 3 biomedicines-10-01910-f003:**
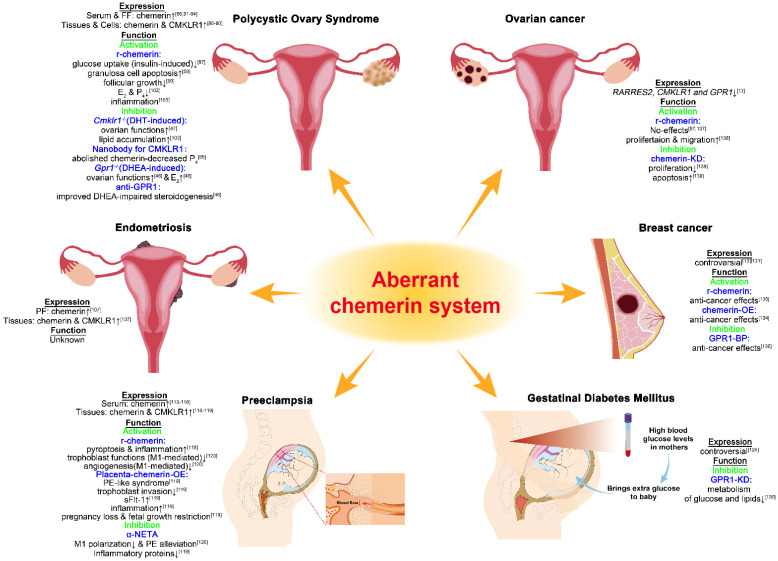
The roles of the chemerin system in reproductive system diseases. The emerging knowledge about the roles of the chemerin system in reproductive system diseases, including PCOS, endometriosis, preeclampsia, GDM, breast cancer, and ovary cancer. Up-regulated levels of chemerin were found in PCOS (serum, FF, and tissues), endometriosis (PF and tissues), and preeclampsia (serum and tissues), indicating a potential diagnostic value for clinical application. The expression status of the chemerin system in GDM, breast cancer, and ovarian cancer requires further verification. Inhibition of CMKLR1 showed potential therapeutic effects for PCOS (*Cmklr1*^−/−^ and nanobodies) and preeclampsia (α-NETA). Inhibition of GPR1 showed potential therapeutic effects for PCOS (*Gpr1*^−/−^ and antibodies), GDM (shRNA), and breast cancer (GPR1-BP). r-chemerin or chemerin-OE displayed a potential anti-breast cancer effect. FF, follicular fluids; PF, peritoneal fluids; r-chemerin, recombinant chemerin. KD, knockdown; OE, overexpression; GPR1-BP, GPR1-binding peptide.

## Data Availability

Not applicable.

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
