# Peer review of "Chemerin: A Functional Adipokine in Reproductive Health and Diseases"

_biomedicines, 2022, doi:10.3390/biomedicines10081910_

Round 1

Reviewer 1 Report

It is a comprehensive and well-written review on the role of chemerin, one of the extensively studied adipokines in recent years in the context of civilizational diseases related mainly to increasing prevalence of obesity. This narrative review focuses on the description of mechanism of action and currently known roles of chemerin in reproduction and reproductive diseases including polycystic ovary syndrome (PCOS), endometriosis and endometritis, preeclampsia, gestational diabetes mellitus (GDM) and gynecological cancers. Finally the authors discuss the limitations nad gaps in evidence related to topic. The manuscript covers a wide range of current literature and is a missing position among reviews. Most of other reviews focused on single reproductive diseases, mainly PCOS and GDM. II have only one comment - figure 1 and 2 are not fully visible in the pdf and they should also present the direction of action (blockage or activation). Apart of that I think the manuscript merits publication in the journal. 

Reviewer 2 Report

Recently many reviews were published on adipokines. While it is a new research direction it is important to summarise this knowledge from time to time.

-line 63- give citations

-line 79- give citations

-line 99 -give citations

-lines 519-520 - provide more details on how chemerin is used for diagnosis/treatment of diseases

-for ERK, IGF, P4, E1, E2, T,  P4, A4 etc. give full names when mentioned for the first time

-give more information on how chemerin interacts with other adipokines in health and disease

Author Response

This manuscript is a resubmission of an earlier submission. The following is a list of the peer review reports and author responses from that submission.